# The Emergence and Decennary Distribution of Clade 2.3.4.4 HPAI H5Nx

**DOI:** 10.3390/microorganisms7060156

**Published:** 2019-05-29

**Authors:** Khristine Joy C. Antigua, Won-Suk Choi, Yun Hee Baek, Min-Suk Song

**Affiliations:** College of Medicine and Medical Research Institute, Chungbuk National University, Cheongju, Chungbuk 28644, Korea; tineantigua@gmail.com (K.J.C.A.); tuckgirlee@naver.com (W.-S.C.); microuni@chungbuk.ac.kr (Y.H.B.)

**Keywords:** avian influenza, dissemination, evolution, epidemiology, avian, H5Nx

## Abstract

Reassortment events among influenza viruses occur naturally and may lead to the development of new and different subtypes which often ignite the possibility of an influenza outbreak. Between 2008 and 2010, highly pathogenic avian influenza (HPAI) H5 of the N1 subtype from the A/goose/Guangdong/1/96-like (Gs/GD) lineage generated novel reassortants by introducing other neuraminidase (NA) subtypes reported to cause most outbreaks in poultry. With the extensive divergence of the H5 hemagglutinin (HA) sequences of documented viruses, the WHO/FAO/OIE H5 Evolutionary Working Group clustered these viruses into a systematic and unified nomenclature of clade 2.3.4.4 currently known as “H5Nx” viruses. The rapid emergence and circulation of these viruses, namely, H5N2, H5N3, H5N5, H5N6, H5N8, and the regenerated H5N1, are of great concern based on their pandemic potential. Knowing the evolution and emergence of these novel reassortants helps to better understand their complex nature. The eruption of reports of each H5Nx reassortant through time demonstrates that it could persist beyond its usual seasonal activity, intensifying the possibility of these emerging viruses’ pandemic potential. This review paper provides an overview of the emergence of each novel HPAI H5Nx virus as well as its current epidemiological distribution.

## 1. Introduction

Wild aquatic birds have always been one of the recognized reservoirs and natural hosts of low pathogenic avian influenza (LPAI) viruses, and upon their circulation in domestic poultry, highly pathogenic variants of avian influenza viruses (AIVs) evolve [1,2,3,4,5,6,7,8]. The first reported case of highly pathogenic avian influenza (HPAI) viruses in domestic fowl was the 1878 Fowl Plague involving chickens in northern Italy [3,5,9]; since that plague, researchers’ perspectives of the biology of HPAI viruses—which emerged from LPAI viruses—have changed. One of the most potent viruses resulting from these biological changes was the HPAI H5N1 virus (A/goose/Guangdong/1/96, or Gs/GD), first isolated from a domestic goose in Guangdong Province, China in 1996 [2,9,10,11,12].

Moreover, this HPAI H5N1 Gs/GD lineage has developed multiple sublineages and has undergone reassortments with other AIVs [1,2,5]. HPAI H5N1 Gs/GD was initially restricted to southern China before it began spreading throughout Asia, Europe, the Middle East, and Africa in 2005–2006 [2,9,10,11,12,13]. As the spread of the virus continued and surveillance increased, more isolates were discovered and analyzed. With the rising number of recorded disease outbreaks in poultry and humans, the WHO/OIE/FAO H5N1 Evolution Working Group defined a specific nomenclature system to group H5 viruses into clades [14,15,16,17]. The criteria mainly involve the following three requirements:Viruses sharing a common node in the phylogenetic tree.Monophyletic grouping with a bootstrap value of ≥60 at the clade-defining node (after 1000 neighbor-joining bootstrap replicates).Average percentage pairwise nucleotide distances between and within clades of >1.5% and <1.5%, respectively.

Primarily, the genetic evolution and antigenic drift of HPAI H5N1 viruses have caused divergence, resulting in the generation of 10 distinct clades [2,9,18]. The major clade 0—which is primarily composed of influenza H5 progenitor viruses—is the closest to the Gs/GD lineage [19,20]. Most of these isolates were observed in Hong Kong and China in 1996. Clades began to split when one isolate did not meet the nucleotide divergence criteria and rather belonged to a discrete monophyletic grouping. As a second-order clade evolves and reaches a similar level of genetic diversity, it can split into third-order clades [20]. From 2001 to 2006, the four clades previously classified were further expanded into 10 distinct clades (0–9). Later in 2006, clade 2 further expanded into five second-order clades [20]. Clade 2 has been observed as a distinct clade which has greatly expanded because of the increased number of isolates. Clade 2.1 includes isolates from Indonesia observed in both avian and human hosts from 2003 to 2007. Clade 2.2 has been isolated from birds and humans from Eastern and Western Europe, the Middle East, and Africa, including the Qinghai Lake Outbreak and Mongolia cases from 2005 to 2007 [19,20,21]. On the other hand, clade 2.3 includes Gs/GD lineage isolates which have caused avian and human influenza virus in China, Hong Kong, Vietnam, Thailand, Laos, and Malaysia from 2003 to 2006 [21]. Clade 2.4 contains avian virus isolates from Yunnan and Guangxi Provinces of China from 2002 to 2005 [17]. Lastly, clade 2.5 includes avian virus isolates from Korea, Japan, and China from 2003 to 2004 and isolates from Shantou, China in 2006 [17,19,20].

The expansion continued as the virus spread across Central Asia and the Middle East to Africa in 2007. By 2008, the second-order clades expanded into third-order clades with the development of 2.1.1–2.3.4 [19]. Interestingly, this expansion was followed by a period with fewer outbreaks from 2009 to 2013 [2]. Unknowingly, during this timeframe, HPAI H5N1 viruses of the 2.3.4 clade reassorted with other influenza viruses resulting in geographical spread [2,5,20]. By 2011, first-, second-, and third-order clades had expanded into additional second-, third-, and fourth-order clades [15,16,17]. Along with the expansion of these clades, previously circulating clades of H5N1 had not resurfaced nor been detected for several years. Since 2008, the isolates or similar isolates belonging to the following clades have been inactive: 0, 2.1.1, 2.1.2, 2.3.1, 2.3.3, 2.4, 2.6, 3, 4, 5, 6, 8, and 9 [20]. These viruses could have been displaced by new clades, as they appear to be extinct [22].

## 2. Emergence of H5Nx

HPAI H5N1 Gs/GD continues to cause outbreaks in poultry in various countries [2,18,23]. Moreover, even with increasing diversity among its internal viral gene, there was no evidence of reassortment of its hemagglutinin (HA) and neuraminidase (NA) genes prior to 2008 [24,25,26]. However, the long stable NA–N1 subtype of this Gs/GD lineage began to switch for several new NA subtypes [2,7,9,19]. This has been elucidated with the isolation of HPAI subtypes H5N5, H5N2, and H5N8, which were shown to bear the same Gs/GD lineage backbone of the H5 clade from domestic ducks and other poultry during live-bird market surveillance in China [2,7,9]. Usually, most H5 viruses of the Gs/GD lineage derive their NA genes from Gs/GD-like viruses or some other N1 origin. However, due to naturally extensive genetic reassortment activity within HPAI H5 viruses and/or several contributing factors (e.g., presence of wild migratory fowls and flyways, duck densities, and poultry farming practices including distribution of vaccines), novel HPAI H5Nx viruses bearing unique NA proteins (i.e., H5N2, H5N3, H5N6, and H5N8) have risen [1,2,8,9,18,22,24,27,28,29,30,31,32,33,34,35,36]. In addition, the extensive divergence of the H5 HA sequences of viruses has warranted clustering the viruses into a systematic and unified nomenclature. On 12 January 2015, the WHO/FAO/OIE H5 Evolution Working Group was prompted to update the H5 hemagglutinin clade nomenclature due to the increased emergence of multiple reassortant viruses since 2012 [14,15,17]. With careful analysis, the WHO/FAO/OIE H5 Evolution Working Group has designated clade 2.3.4.4 as the unified classification for H5Nx viruses—H5N2, H5N5, H5N6, H5N8, and 2.3.4.4 H5N1 virus subtypes [14,15,17].

The group of viruses sharing this common backbone lineage have been tagged as the H5Nx viruses, which are currently causing public health threats with a recurrent number of sporadic to enzootic outbreaks in various parts of the globe [2,7,9]. Distinctly, the HA cleavage site is one of the key indicators of influenza viruses in terms of systemic replication and lethal infection [37]. Usually viruses would possess a single arginine on its cleavage site, indicating low pathogenicity [38]. All H5Nx viruses share the consensus multibasic amino acid sequence of -RRRKR/G- at the HA cleavage site between HA1 and HA2, suggesting that these subtypes are highly pathogenic [12,24,25,37,38,39,40,41,42,43,44,45,46,47,48].

In terms of molecular characterization, members of clade 2.3.4.4 are highly similar and related to the HA protein of the HPAI clade 2.3.4 A/wild duck/Hunan/211/2005. Moreover, De Vries et al. found that seven amino acid positions, namely, K86R, T160A, N187D, K222Q, S227R, N244H, and A267T, are maintained in all 2.3.4.4 member viruses [49]. They also discovered two substitutions that are unique only to clade 2.3.4.4—K222Q and S227R. Uniquely, these substitutions were not previously observed in any HPAI H5N1 viruses [26,49,50,51].

## 3. Evolution of H5Nx Subtypes

By the end of 2008, HPAI H5Nx viruses of clade 2.3.4.4 began yielding novel reassortant viruses (e.g., H5N2, H5N3, H5N6, H5N5, and H5N8) [2,9,22,34,36]. These viruses all share the same H5 monophyletic group with the same HA gene as H5N1 viruses. This group of recently emerged HPAIs has caused frequent and widespread epizootic outbreaks in various parts of Asia, particularly in China and Southeast Asian countries [2,52]. Aside from the economic losses within the poultry industry, this group of viruses not only poses a threat to animal health but also continuously raises concerns for public health. Therefore, understanding the evolution and emergence of these potentially pandemic strains is of great importance.

### 3.1. H5N5

Domestic ducks, among poultry, are highly regarded as the reassortant vessel for most avian influenza viruses [46,53]. Furthermore, the extensive prevalence of live poultry markets in China enhances the reassortment activities of influenza subtypes [24,52,54]. The combination of these two factors has contributed significantly to the formation of the first H5Nx virus [12,46,48,53]. It was in the latter part of 2008 when the first H5N5 virus was isolated in China (Figure 1) [46,53].

Specifically, the first reassortant subtype was isolated from apparently healthy mallard ducks (*Anas platyrhynchos*) in a live poultry market surveillance program [12,46,48,53,55,56]. Two isolates, A/duck/eastern/China/008/2008 and A/duck/eastern/China/031/2009 were detected in live poultry markets in eastern China [12]. Furthermore, three more H5N5 isolates were isolated from ducks during the same period of live poultry market surveillance in Guangdong during the same year [12,46]. In 2010, another case was isolated from ducks in Central China [48].

H5N5 are multiple reassortants of possible parent viruses: H5N1 A/duck/Eastern-China/108/2008, H5N1 A/duck/Eastern-China/909/2009, and H6N5 A/duck/Yangzhou/013/2008. Gu et al. (2011) further speculated that H5N5 viruses are contemporary reassortants of Eurasian subtype H5N1, some unidentified subtype, and/or the H6N5 avian influenza virus [53]. Conversely, Zou et al. (2012) theorized that H5N5 is a result of multiple reassortments of the H10N5 and H5N1 viruses, which have been circulating in China since 1983 and 1997, respectively, and with another circulating AIV in China [48].

Although the H5N5 subtype was the first H5Nx reported, there are only a few documented records of H5N5 viruses of the 2.3.4.4 clade. Furthermore, no records of human infection nor massive poultry outbreak have been reported [12]. Moreover, this subtype, from the time of its detection, was rarely isolated [48]. According to the Animal and Plant Health Agency (APHA)-UK, during 2016–2017, H5N5 re-emerged because H5N8, which had been predominantly circulating worldwide, lost its N8 during its reassortment activities with LPAI viruses from circulating wild birds and this led to the incorporation of the N5 gene [13,57]. Based on the records of the Food and Agriculture Organization (FAO)’s Empress-I, aside from its detection in China, H5N5 has only 18 other noted occurrences in 11 European countries, namely, France, Italy, Montenegro, Croatia, Poland, Czech Republic, Greece, Germany, Serbia, Slovenia, and the Netherlands, between 2016 and 2017 [58].

### 3.2. H5N8

With the massive avian influenza surveillance and sample collection among domestic live poultry markets and farms in China, the novel subtype H5N8 (A/duck/Jiangsu/k1203/2010), along with the three H5N5 subtypes, was first detected in eastern China in 2010 (Figure 1) [46].

Zhou et al. (2013) have also revealed that the five H5N8 internal genes (PB1, PB2, PA, M, and NS) share high sequence homologies with H5N1 isolated in eastern China from 2005 to 2006 [46]. Nonetheless, as for the NP gene, it was observed to have high sequence similarities with H6N2 viruses isolated from Guangdong Province during 2001–2003. Thus, it suggests that H5 HPAI bearing the N8 gene may have originated in China [47].

Furthermore, Ma et al. (2018) reported that the maximum clade credibility (MCC) trees generated for HA and NA genes of the first H5N8 isolate revealed that the reassortment likely happened in August 2009 [41]. Their study further inferred that reassortment happened between domestic poultry “HPAI H5Ny and HxN8” and circulating wild aquatic birds [41,47]. Known as one of the most prevalently circulating HPAI in China in 2009, H3N8 viruses were mainly identified as the subtype with the highest probability of being the neuraminidase donor [41,46]. Thus, it was theorized that this reassortment which occurred between domestic poultry and wild aquatic birds likely paved the way for the formation of the novel H5N8 [41].

When H5N8 was first detected and isolated, it was from apparently healthy ducks (*Annas platyrhynchos*) in the province of Jiangsu [39,41,46]. No H5N8 outbreaks were recorded in China during that time, and it was first isolated and reported in 2013. However, an outbreak caused by H5N8 in poultry and wild birds was reported in the Republic of Korea from January 2014 to July 2015. The detected H5N8 in Korea is highly homologous with the same virus isolated during the live poultry market surveillance program in China [41]. Japan also had the same outbreak in April 2014 [13]. Moreover, with the aid of the migratory pathways, in late 2014, H5N8 began spreading explosively in Russia and six other European countries (i.e., Germany, Hungary, Italy, the Netherlands, Sweden, and the United Kingdom) [9,13,29] in early 2015. The spread of HPAIV (H5N1) from Asia to Europe may have been brought about by several factors, such as transport, production, or wild bird migration [59]. Some scientific papers have separated the sublineages of H5N8 and referred to each group as either as 2.3.4.4 “Buan-like” or “Gochang-like” [13,60].

H5N8 Buan-like sublineages, or Group A, are the viruses detected which were established and first detected in Europe and some Asian countries in 2014. Buan-like H5N8 precursor viruses were believed to have been initially detected in eastern China before their intercontinental spread [13,46,60,61]. Although outbreaks did not persist for a long period, this was the first incursion of clade 2.3.4.4 of Gs/GD lineage into Europe [13,62]).

In 2016, the second wave of H5N8 was detected. This group of H5N8 was then called “Gochang-like” viruses [13]. This novel H5N8 reassortant was detected in a wild bird in Qinghai, China in May 2016, before it persisted in intercontinental spread [9,29,63]. Furthermore, Lee et al. reported that this H5N8 HPAI virus group, which had also been detected in the Tyva Republic, Russia, is different from the H5N8 group, which persisted in Europe in 2014 [60]. Moreover, in their study, they showed that this reassortant has three out of eight genes as in the case of the Gochang-like viruses. The reassortant is a product of Eurasian lineage LPAI and HPAI H5N8, containing five Eurasian LPAI segments (PB2, PB1, PA, NP, and M) [9].

Since the re-emergence of the novel reassortant H5N8 in 2016, a wide range of avian species have been reported to be affected by H5N8 HPAI viruses in 49 countries [13,58,64]. Studies revealed that this new novel reassortant was then disseminated to Siberia during migratory fowl migration [2,13,27]. In 2016, H5N8 viruses continued to spread across Europe, Asia, and Africa [29,64]. With the contingencies made in animal health regulations, poultry trade, and country quarantine procedures, the number of outbreaks has gradually declined. However, even with the combined strategies in Asia, H5N8 still persists, especially in China [2,18,38,39,41,43,46,47,54,63,65,66].

Based on data collated from the Empress-I database run by the FAO, there were already almost 3,400 reports globally of HPAI H5N8 infection, alone and/or combined with other HPAI viruses from 2013 to 2018 [58]. Although this subtype has not been reported to cause human infection, the continuous spread of this virus is alarming and should be regarded with precautionary measures. The circulation of this novel subtype from domestic poultry to wild birds has generated novel reassortants which have spread globally [39,41,43,54]. Moreover, Ma et al. (2018) reported that in the process of adaptation, H5N8 amino acids changed gradually during its transmission from one host to another [41]. Recently, the rapid adaptation of the virus in mammalian hosts was observed; the mouse-adapted virus significantly increased virulence in mice, further highlighting potential concerns for public health [67].

### 3.3. H5N6

After the emergence of H5N5 viruses in clade 2.3.4.4, their HA gene was observed to be closely associated with the emergence of NA subtypes, with N2, N5, N6, and N8 variants in poultry and wild birds [68]. The revolution of H5Nx viruses began in July 2008 when an H5N5 virus of Guangdong Province (A/duck/Guangdong/wy19/2008) spread and caused an outbreak in the northeastern province of Jiangsu [25,52]. In 2010, it was recognized that Jiangsu Province became a central pool of avian influenza viruses transmitting towards the northern and southern parts of China [25]. The interplay and movement of migratory fowls across China has resulted in the development of the highly pathogenic avian influenza virus H5N6 [24,33,52,68,69].

The Chinese government enforced active surveillance and sample collection activities to secure public health. Hence, during the surveillance, it was found that H5N6 viruses have dominated and spread throughout the country [24]. From the time it was first isolated and detected, this virus has continuously persisted in China and caused fatal human infection [24,33,52,68,69]. To date, HPAI H5N6 is one of the few AIV subtypes reported to cause human infection [44]. Understanding the origins and genesis of each virus subtype can help determine their relationship with other subtypes in the same clade.

Previous studies have revealed that H5N6 emerged from a common H5 progenitor strain of the 2.3.4.4. clade reassorted with the neuraminidase of the Eurasian lineage originating from the H6N6 A/duck/Guangxi/2281/2007 [24,25,44,68] However, Yang et al. (2017) revealed that this novel reassortant emerged from two evolutionary pathways [33]. From 2010 to mid-2012, the HA H5N2 of clade 2.3.4.4 reassorted with the NA of H6N6 (A/duck/Guangxi/2281/2007) and further reassorted with the six internal genes of a chicken host H5N1 of clade 2.3.2.1c [33]. The H5N6 reassortant (“Reassortant A”) resulting from this pathway has been circulating in Xinjiang, Jilin, and northern China [24,25,26]. Moreover, in 2013, H5 viruses of the 2.3.4.4 clade have then spread widely to western parts of China, causing outbreaks in Sichuan Province and neighboring countries such as Vietnam and Laos [2,44].

Beyond 2013, H6N6 viruses bearing deletions at the 59–69 position in the stalk regions of its NA have reassorted with the HA of H5N8 of clade 2.3.4.4 and further reassorted with the H5N1 of 2.3.2.1c (“Reassortant B”) [33]. Likewise, Reassortant B viruses have also been detected in China, Vietnam, and Laos. Two years later, these Reassortant B H5N6 viruses then further reassorted with H9N2, generating the H5N6 “Reassortant C”. Moreover, these newly reassorted H5N6 viruses were reported in Yunnan and Guangdong Provinces [26,33].

Apparently, no matter the evolutionary pathway from which these sequential multiple-step reassortant viruses emerged, all were reported to cause human infection [24,25,33]. Based on the data within the FAO’s Empress-I database, there were 19 confirmed cases of human infections due to HPAI H5N6. Notably, all 19 reports were confirmed positive and distributed only in China [58]. Most of the human cases are attributed to Reassortants A or B. However, the first case of H5N6 human infection in 2014 was due to H5N6 Reassortant A [33,69]. Reassortant C viruses were noted to have lower virulence compared with Reassortants A and B [33].

To date, based on the Empress-I database managed by the FAO, H5N6 cases are confined to Asia and Europe. H5N6 spread across Laos and Vietnam in 2014, causing economic losses due to increased poultry mortalities [58]. By 2016, H5N6 was responsible for a series of outbreaks in Asia, including a poultry outbreak in Japan, Myanmar, and the Republic of Korea. Moreover, the highest number of human cases due to H5N6 was recorded in December 2015 (including 9 out of 16 human fatal cases to date) [25,58]. By 2017, H5N6 had already spread through China’s neighboring countries such as Taiwan and the Philippines. Furthermore, it began spreading to some European countries such as Greece, Germany, the Netherlands, and Switzerland. Primarily, migratory birds have been the key player for the expansion and genetic reassortment of HPAIVs, particularly for H5N6 from Asia to Europe [24,25,33,52,68].

### 3.4. H5N2

During the live bird market surveillance of AIVs, the regenerated H5N2 virus was isolated in apparently healthy domestic poultry in eastern China in 2011 [70]. More isolates were found in ducks and geese which were transported from Shandong and Jiangsu Provinces to eastern China for trade. Upon checking, the first isolate H5N2 gene origin was rooted in the HA gene of H5N1 (A/goose/Guangdong/1/1996), whereas its N2 shared the highest homology with the NA gene of the duck H3N2 (A/duck/Eastern China/142/2006) [70,71]. Nonetheless, in 2011, H5N2 outbreaks in Shandong and six more provinces in China including Tibet were reported [28,70].

H5N2 has also been reported in Taiwan. Since December 2014, a series of HPAI H5N2 outbreaks, coupled with H5N8 and H5N3, were reported by the Taiwan government. In January 2015, the occurrence of H5N2 of Eurasian lineage had spread widely in various provinces of Taiwan with numerous accounts of mortalities among poultry and wild birds [2,72]. With the destruction of more than 2 million poultry from more than 950 poultry farms, Taiwan declared this event as the “largest epidemic” of avian influenza in their country’s history [28].

The panzootic spread of H5Nx exhibited its intercontinental transmissibility in 2014 [73,74,75]. H5Nx viruses were introduced to the North American continent through migratory fowls [5,22,27,41,76]. HPAI H5N8 viruses have reassorted with North American LPAI viruses in wild birds resulting in the generation of H5N1 and H5N2 [58]. The generated reassortant HPAI H5N2 contains five segments of the H5N8 and three from the North American LPAIV [5,22,27]. On 28 November 2014, mortalities in an 11,000-head turkey farm in Canada were reported; sequencing confirmed that H5N2 was the causative agent of this outbreak [77]. This was the first time that a Eurasian lineage of HPAI crossed into North America, specifically through Canada [5,77]. Since then, additional cases have been detected on poultry farms and in wild birds. Furthermore, an estimated 50 million birds have been culled and stamped out in numerous areas [7]. Notably, the majority of these outbreaks were deemed to be caused by the H5N2 subtype [5].

### 3.5. H5N3

The detection of the first H5N3 virus of Eurasian lineage was observed during Taiwan’s largest avian influenza epidemic in 2014 [28,72]. During avian influenza surveillance conducted by Taiwanese authorities, two novel HPAI H5N3 viruses from a chicken (a/chicken/Taiwan/a174/2015) and a duck (A/duck/Taiwan/a180/2015) on a farm located in Pingtung Province were detected [28]. Chang et al. (2016) revealed that the nucleotide identities of the representative isolates—three (HA, PA, and NS) likely of H5N3 genes of 99.82–100%—are highly associated with the wild bird H5N8 (A/crane/Kagoshima/KU13/2014) [28]. On the other hand, the N3 gene originated from H10N3 (A/duck/Jiangxi/33629/2013). The PB2 shares 97% similarity with that of the H6N2 isolated from a coot in Georgia (A/common coot/Republic of Georgia/1/2010) [28]. However, along with the H5N8 and H5N2 subtypes, this novel reassortant contributed to the massive outbreak of AIV in Taiwan (Figure 1) [72].

There were very few documentations of this subtype, and its occurrence has also been limited. To date, there have been reports of H5N3 causing outbreaks in China, Germany, and the Netherlands. The cases reported were limited and occurred sporadically.

### 3.6. H5N1

HPAI H5N1 of the Gs/GD lineage has been known for its impact and novel reassortants. It has been widely disseminated around the globe and reported in various hosts. Thus, this subtype has a long history of causing massive outbreaks affecting the poultry industry and poses a continuous threat to public health. According to Ducatez et al. (2017), H5N1 has three large expansions in clades 2.2, 2.3.2.1, and 2.3.4.4 [21]. Having been widely established in various hosts and environments, it was hardly surprising that a reassortant of H5N1 appeared under clade 2.3.4.4 recently.

Between December 2013 and 2014, a wave of HPAI outbreaks occurred in three counties of Yunnan, China. The reported outbreaks in Tonghai, Anning, and Dali killed 20,000 heads of layer poultry [30]. Upon sequence verification, the virus detected and isolated was found to belong to clade 2.3.4.4, sharing the highest identity of its HA gene with the isolates from H5N6 Laos (A/chicken/Laos/LPQ001/2014) and H5N6 China (A/duck/Guangdong/GD01/2014) [30]. Moreover, it was also identified that the NA gene of this emerging H5N1 was closely related to the N1 of isolates of clade 2.3.2, which are A/chicken/Vietnam/NCVD-KA423/2013, A/duck/Hunnan/S4220/2011, and A/duck/Zhejiang/213/2011. Nonetheless, all the other six genes of this novel emergent are shared with H5N2 A/duck/Jiangxi/JXA132023/2013 [30].

### 3.7. H5Nx Provisional Grouping

With the continuous emergence and observed diversity patterns of H5 viruses, Lee et al. proposed a provisional grouping for viruses belonging to clade 2.3.4.4. The H5Nx viruses were grouped into four (A-D) based on the phylogenetic relationships and the temporal evolutionary history of viruses when investigated using molecular clock analysis [10]. Moreover, Lee et al. also demonstrated that the genetic relatedness of viruses in each group are well supported by the observed high bootstrap values (>70%) and long HA branches [9,10]. Group A or 2.3.4.4A is composed of viruses such as the following: a) H5N8, identified from China in early 2014 and from South Korea, Japan, Taiwan, Canada, European countries and North America; b) H5N1 and H5N2 from North America; and c) H5N2 and H5N3 from Taiwan. On the other hand, group B or 2.3.4.4B is primarily composed of H5N8 viruses identified in China in 2013 and South Korea in 2014 [9,10]. Moreover, recent studies have suggested that H5N6 from South Korea in 2017–2018 as well as H5N5 and H5N8 from Germany in 2016–2017 also belong to this group by phylogenetic relatedness [78,79]. Group C of 2.3.4.4 clade is composed of H5N6 from China and Laos in 2013–2014 and H5N1 identified from Vietnam and China in 2014 [9,10]. Kim et al. also reported that H5N6 identified in Korea in 2016–2017 shares the phylogenetic relatedness in this group of 2.3.4.4 [80]. Remarkedly, 2013–2014 H5N6 identified in China and Vietnam were grouped in 2.3.4.4D [9,10]. Although this provisional grouping remained unofficial, few studies have adapted the groupings to elucidate the diversity patterns of clade 2.3.4.4 viruses or H5Nx based on molecular clock analysis.

## 4. Evaluating the Decennary Distribution of H5Nx

H5Nx viruses have made a significant impact, with damage to the economy and trade as well as their imminent threat to public health. Since their first emergence to the present, H5Nx viruses have spread widely in various countries and territories around the world. Based on the records found in the FAO’s Empress-I database, HPAI H5Nx have caused at least one outbreak in poultry and/or wild fowls in 61 countries or territories [58].

According to the OIE Influenza Situation Report (August 2018) and the World Animal Health Information Database (WAHIS) Interface, there have been 12 different highly pathogenic avian influenza subtypes reported since 2013 worldwide. Moreover, 6 out of these 12 reported AIV subtypes belong to clade 2.3.4.4. H5Nx. In terms of regional distribution, H5Nx viruses were highly distributed among Asian and European regions (Table 1) [81]. On the other hand, there were three subtypes which have been reported in the American region, namely, the novel reassortants H5N1, H5N2, and H5N8, whereas the African region has recently reported the incursion of H5N8 affecting both domestic poultry and wild birds. Among the clade 2.3.4.4 subtypes recorded, interestingly, H5N8 were widely distributed in almost all regions. Nonetheless, there were no reports recorded of H5Nx occurrence in and incursion into the Oceania and Antarctic regions.

In Figure 2A–F, the map demonstrates the distribution of reports of recorded H5Nx outbreaks from 2008 to 2018. The unpinned areas represent the regions and countries which have no reports nor detection of the HPAI H5Nx viruses based on the FAO’s Empress-I data from the specified time frame. Overall, based on the map generated, H5Nx has been detected in almost half of the countries in the Asia-Pacific region. Out of the 48 countries in that region, 18 (18/48) countries have reported at least one confirmed case of H5Nx subtype infection in avian species. These countries include Cambodia, China, Cyprus, Hong Kong, India, Iran, Iraq, Israel, Japan, Kazakhstan, Kuwait, Laos, Myanmar, Nepal, Philippines, Republic of Korea, Saudi Arabia, Taiwan, and Vietnam. Among these countries, China, Korea, and Taiwan had the most reported cases of HPAI H5Nx infection from 2008 to 2018 [58].

Furthermore, it can also be seen that H5Nx viruses have been confined to certain areas of North America. No reports of H5Nx were recorded from Central and South America. European and African countries have also confirmed H5Nx occurrences, while it remains undetected in Oceania and Antarctic regions (Figure 2A–F). The European region consists of 44 different countries, which include transcontinental countries such as Russia, Azerbaijan, Georgia, and Turkey. Out of these countries in the European region, 31 out of the 44 countries have reported at least one confirmed report by one or multiple H5Nx subtypes. Interestingly, based on the FAO’s Empress-I collated data, all 31 countries, except for Montenegro, have reported H5N8 subtype infection.

In terms of the H5Nx map distribution, H5Nx was initially reported in Asia, specifically in China prior to 2014 (Figure 2A). Neighboring countries such as Korea, Japan, Laos, and Vietnam also reported outbreaks of H5Nx avian influenza. European and American continents have reported H5Nx incursion in 2014 (Figure 2B). Subsequently, a series of outbreaks have been observed intra- and intercontinentally in 2015 (Figure 2C).

In 2015, an abrupt spike in the number of H5N2 cases occurred in North America (Figure 2C). Based on the FAO’s Empress-I records, the reported cases resulted in a cumulative loss of 22.8 million poultry birds (Figure 3) [58]. Based on the complex nature of HPAI viruses, different subtypes emerged and caused outbreaks the following year. With efficient biosecurity, early detection, and stringent control measures, the risk of poultry outbreak can be minimized [2,13,22]. This has been observed with the abrupt lack of continued circulation of H5N2 in North America (Figure 2C–F) [9,10,58]. Nonetheless, H5N6 displaced H5N2 in 2016 and a slight decline in poultry mortalities was observed. HPAI H5N6 outbreaks in 2016 were solely confined to Asia, specifically, the 2016 outbreak of H5N6 in the Republic of Korea, which included a recorded poultry mortality loss of 21.3 million heads [58]. Moreover, outbreak cases were also observed in China, Taiwan, Myanmar, and Hong Kong [13,58].

H5Nx of Gs/GD lineage reached the African continent in 2016 (Figure 2D). Prior to 2016, there were no H5Nx cases reported in the region. Specifically, it was in November 2016 when Egypt and Tunisia first reported the first detection of the HPAI H5N8 virus. This was immediately followed by the eruption of reports from Nigeria, Niger, Cameroon, and Uganda. It has long been suggested by many studies that one of the major risk factors for the extensive spread of the virus is the seasonal migration of wild aquatic birds [2,5,13,22,29,32,64,82]. Combining wild bird migration and the favorable environmental condition of African wetlands, it is no wonder that within the span of two years, there have been 285 reports of HPAI H5N8 in domestic poultry as well as marine bird populations in Africa [64].

The occurrence of H5N5, H5N3, and H5N1 of clade 2.3.4.4 was limited. H5N3 has been limited to its emergence in the Taiwan outbreak and China [28,72]. H5N1 of clade 2.3.4.4 was characterized by its sporadic and mixed occurrence with other subtypes in Vietnam and China [58]. Outbreaks involving multiple AIVs were mostly observed in Asia, specifically in Taiwan and China. Furthermore, H5N5 reoccurred in 2016–2017, with outbreaks in 11 countries of the European region; however, the associated losses were minimal compared with those caused by H5N8 or H5N2 [58].

By 2017, a huge decline in poultry losses was recorded and observed. However, a significant number of HPAI H5N8 and H5N6 viruses were reported and documented in various regions (Figure 2A–F and Figure 3). H5N8 has greatly affected the European region, with 1380 outbreak cases tied to a cumulative poultry loss of 3.7 million birds (Figure 2E and Figure 3). In addition, the Republic of Korea has incurred 6.7 million poultry losses from outbreaks of both subtypes. There were also reports of H5N6 outbreaks in the Philippines, China, and Saudi Arabia, causing the death and destruction of an estimated 4 million birds.

In 2018, H5N2, H5N6, and H5N8 have recorded a cumulative total of 6.98 million poultry losses in Africa, Europe, and Asia (Figure 2F and Figure 3). Most of these poultry losses were due to outbreaks in Asian countries. Specifically, based on the FAO’s Empress-I database, H5N6 and H5N8 have greatly contributed to 1.2 and 5.1 million poultry losses, respectively. There were also sporadic cases of H5N2 recorded from Taiwan and H5N8 from Nigeria and South Africa (Figure 2D–F and Figure 3). Although the total number of H5Nx outbreaks has decreased in 2018, these continual records of outbreaks suggest the continuous persistence of H5Nx viruses, with the probable risk of re-emergence.

The divergence brought about by clade 2.3.4.4 is highlighted with the emergence of novel reassortants from 2008 to 2018. These reassortants had an immense impact on the poultry industry of affected countries. Since 2008, detection and isolation of various H5Nx viruses from poultry farms and migratory fowls have already been recorded. There have been records of outbreaks in various regions, but they have been rather sporadic. Other factors such as poultry and duck densities as well as farming practices have aggravated the occurrence and continuous persistence of H5Nx outbreaks in those reported countries. However, wild bird migration has been primarily speculated to have most greatly impacted the incursion of the Eurasian lineage clade 2.3.4.4 into the four mentioned regions.

## 5. Conclusions

The rapid emergence and circulation of clade 2.3.4.4 H5Nx viruses (H5N2, H5N3, H5N5, H5N6, H5N8, and H5N1) are of great concern, as they continuously impose a public health threat. A better understanding of the evolution and emergence of these novel reassortants helps better elucidate their complex nature. Primarily, H5Nx has high inter- and intracontinental transmissibility, as evidenced by its global spread. The eruption of reports of each H5Nx reassortant through time demonstrates that it could persist beyond its usual seasonal activity, intensifying the possibility of these emerging viruses’ pandemic potential.

## Figures and Tables

**Figure 1 microorganisms-07-00156-f001:**
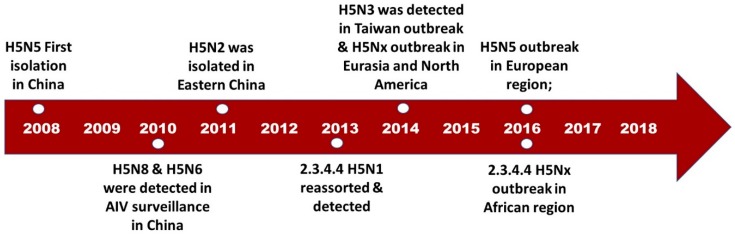
H5Nx timeline of evolution and continuous emergence (2008–2018).

**Figure 2 microorganisms-07-00156-f002:**
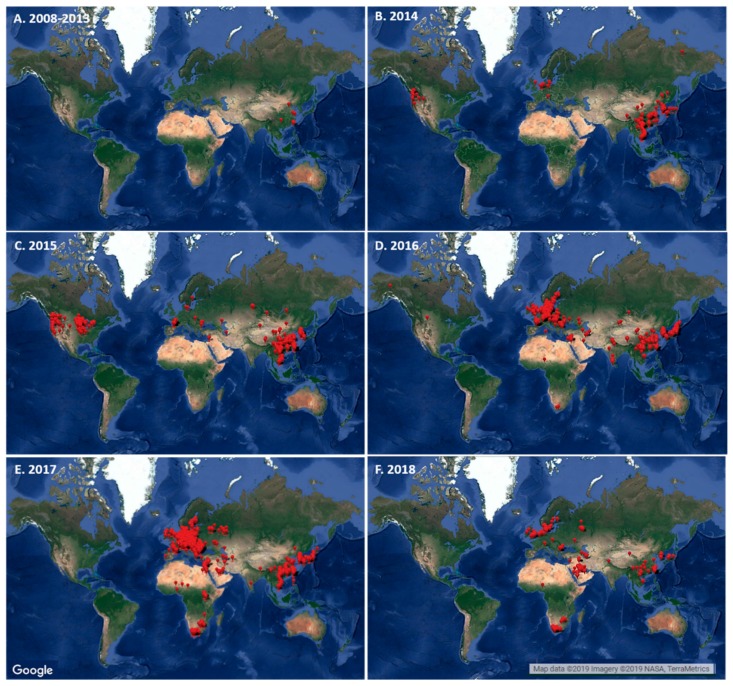
Distribution of reported H5Nx occurrences in various regions per specific period: (**A**) 2008–2013, (**B**) 2014, (**C**) 2015, (**D**) 2016, (**E**) 2017, and (**F**) 2018. Red pins in the map denotes a record of confirmed reported occurrence of H5Nx in the specified area. The map was generated using the collated reports recorded in the database of the Food and Agriculture Organization (FAO)’s Global Animal Disease Information System, Empress-I (http://empres-i.fao.org/eipws3g/) and was mapped using Maptive^®^ (https://www.maptive.com/), an online mapping software utilizing Google Maps.

**Figure 3 microorganisms-07-00156-f003:**
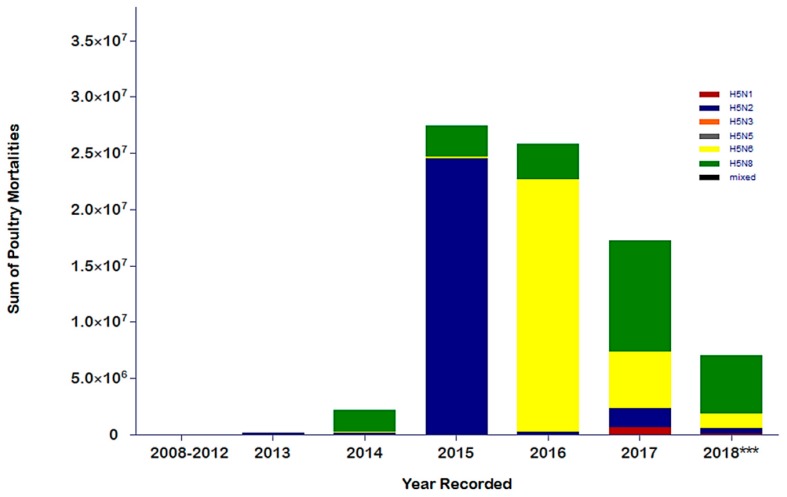
Epi curve of the cumulative sum of mortalities ^a^ recorded due to H5Nx from 2008 to June 2018. The data were generated based on the downloaded data from the FAO’s Global Animal Disease Information System, Empress-I (http://empres-i.fao.org/eipws3g/) [58]. *** For 2018, it should be noted that some of the poultry mortalities from the African region in 2018 were not all reflected from the collated Empress-I data. ^a^ Cumulative poultry mortalities is the total sum of all the mortalities including the number of dead, slaughtered, and destroyed due to the specified disease cause during the specified period.

**Table 1 microorganisms-07-00156-t001:** Regional distribution of clade 2.3.4.4 highly pathogenic avian influenza (HPAI) H5Nx subtypes from 2008 to 2018.

Region	H5Nx Subtype Present
Africa	H5N8
Americas	H5N1, H5N2, H5N8
Asia	H5N1, H5N2, H5N3, H5N5, H5N6, H5N8
Europe	H5N1, H5N2, H5N5, H5N6, H5N8
Oceania and Antarctic	none

Tabulations were generated based on the data available in the OIE Situation Report for Highly Pathogenic Avian Influenza (August 2018), the OIE World Animal Health Information Database (WAHIS) Interface (http://www.oie.int/wahis_2/public/wahid.php/Wahidhome/Home), and the FAO’s Global Animal Disease Information System, Empress-I (http://empres-i.fao.org/eipws3g/).

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
