# Peer review of "The Emergence and Decennary Distribution of Clade 2.3.4.4 HPAI H5Nx"

_microorganisms, 2019, doi:10.3390/microorganisms7060156_

Reviewer 1 Report

This manuscript entitled “The Emergence and Decennary Distribution of Clade 2.3.4.4 HPAI H5NX” is a review article focusing on HPAI H5NX viruses of HA 2.3.4.4 clade. The authors summarized the literature on these viruses’ emergence, genetic evolution per subtype, and geographical distribution over the last decade.

The review is of interest and well written but I have the following comments and suggestions:

1.       In the Introduction section: Gs/GD HPAI H5N1 emerged in Europe, the Middle East and Africa in 2005-2006 and not 2003.

2.       1st and 2nd line from the bottom page 2: 2 sets of references are listed in the same sentence likely by mistake?

3.       References numbers have been shifted at some point (or some are inaccurate), please revise throughout. A few (non exhaustive/comprehensive) examples: reference 69 should be 68 at the very end of the 3.2 section; section 3.5 after Fig 1 it should not be reference 74; bottom of page 9: reference 66 should read 65; legend of Figure 3: it should not be reference 34, etc. This problem makes it quite difficult to cross-check the literature and to review the present manuscript.

4.       Figure 2: I wonder how the authors dealt with clade 2.3.4.4 H5N1 viruses on the maps. I think they should be included in the analysis but to my knowledge there is no option on the EMPRES-I map to include the 2.3.4.4 and not the non 2.3.4.4 per subtype (one can just pick HP/LP H5N1, H5N2, etc) It looks like they were not selected here (the HPAI H5N1 outbreak in Côte d’Ivoire and Togo in 2016-2017, clade 2.3.2.1c are for example not on the map), please indicate this in the legend. Similarly, non 2.3.4.4 H5Nx seem to have been included on the map (the European 2015 viruses were for example non 2.3.4.4 (non Gs/GD) but they are indicated on the map): please adjust and correct Figure 2.

5.       4 lines before Fig 3: to my understanding there is no link between a number of outbreaks and a pandemic potential, please rephrase.

6.       A couple of English mistakes should be corrected: examples: an extra “that” to delete 3rd line of section 3; Cameroon instead of Cameron

Author Response

Dear Reviewers:

We highly appreciate the reviews and comments our manuscript, entitled: “The Emergence and Decennary distribution of clade 2.3.4.4 HPAI H5Nx” have received. This manuscript was conceptualized on the idea of understanding the diversity and continuous emergence of 2.3.4.4 H5Nx viruses. 

Hence, we would like also to sincerely send our utmost gratitude to you as our reviewer, for the comments and suggestions given for the improvement of this paper. We have carefully re-checked and re-analyzed all the information indicated in this manuscript (maps, references and noted and added information). We hope that as we include our point by point responses for each comment/suggestions/clarification, we could clear and clarify all the information regarding this manuscript.

Thank you very much.

Best regards:

Khristine Joy Antigua

Comment/suggestion/Clarification

1.      Regarding References:

A.     References numbers have been shifted at some point (or some are inaccurate), please revise throughout. A few (non-exhaustive/comprehensive) examples: reference 69 should be 68 at the very end of the 3.2 section; section 3.5 after Fig 1 it should not be reference 74; bottom of page 9: reference 66 should read 65; legend of Figure 3: it should not be reference 34, etc. This problem makes it quite difficult to cross-check the literature and to review the present manuscript.

B.      1st and 2nd line from the bottom page 2: 2 sets of references are listed in the same sentence likely by mistake?

Response: We stand corrected for the errors which may have occurred in the process of updating of the reference library. We have re-checked the whole document and now all has been corrected. All references were counter-checked for its position and correct citation. 

2.      For Validation:

a.      In the Introduction section: Gs/GD HPAI H5N1 emerged in Europe, the Middle East and Africa in 2005-2006 and not 2003.

Response: We have counter-checked also the suggested information and we stand corrected. The line now reads as: “HPAI H5N1 Gs/GD was initially restricted to Southern China before it began spreading throughout Asia, Europe, the Middle East, and Africa in 2005-2006.”

b.      A couple of English mistakes should be corrected: examples: an extra “that” to delete 3rd line of section 3; Cameroon instead of Cameron

Response: We have checked the cited mistakes, and have it corrected. We have also checked and re-read the whole manuscript for re-checking.

 c.       4 lines before Fig 3: to my understanding there is no link between a number of outbreaks and a pandemic potential, please rephrase.

Response: We understand the point of the reviewer and we have rephrased the statement. It now reads as: Although the total number of H5NX outbreaks has decreased in 2018, these continual records of outbreaks suggest evidences of H5NX viruses continuous persistence with probability risk of re-emergence.

d.      Figure 2: I wonder how the authors dealt with clade 2.3.4.4 H5N1 viruses on the maps. I think they should be included in the analysis but to my knowledge there is no option on the EMPRES-I map to include the 2.3.4.4 and not the non 2.3.4.4 per subtype (one can just pick HP/LP H5N1, H5N2, etc) It looks like they were not selected here (the HPAI H5N1 outbreak in Côte d’Ivoire and Togo in 2016-2017, clade 2.3.2.1c are for example not on the map), please indicate this in the legend. Similarly, non 2.3.4.4 H5Nx seem to have been included on the map (the European 2015 viruses were for example non 2.3.4.4 (non Gs/GD) but they are indicated on the map): please adjust and correct Figure 2.

Response: We understand the point of the reviewer. To generate each map, we reviewed the data recorded in EMPRESS-I database and counter checking was done for each case country occurrences per year with the available information that agrees with the collated references in this manuscript. We also countercheck the clade classification of H5N1 or H5N2 if the information provided matches with those submitted genomic sequences available in the Influenza Research database. We stand corrected for the non-2.3.4.4 which have been pinned or 2.3.4.4 H5NX that we may have or not have included in the map presented. Hence, to clarify and validate the map generated and clear the misunderstanding from Empress-I Map generation, we have downloaded all the available reports per year group and made new map using another software. Each group subsets of data were counter-checked for the occurrences of subtypes based on the information (location, year, date,) and compare it if there are records available from other databases (IRD, Gisaid, OIE, Pubmed, published journals, web news and articles) for counter-checking of clade validity. After the counterchecking each group subsets, maps were plotted and generated using Maptive® (https://www.maptive.com/), an online mapping software which utilizes Google Map. Now the map is presented as Figure 2.

Reviewer 2 Report

In their manuscript “The Emergence and Decennary Distribution of Clade 2.3.4.4 HPAI H5NX”, Antigua and co-workers review the emergence and epidemiological distribution of Clade 2.3.4.4 H5NX influenza viruses for the past 10 years. The topic is highly valid and interesting. The biggest share in the manuscript is devoted to evolution of clade 2.3.4.4 H5NX subtypes and the rest addressing distribution of viruses but not discussed about the distinct four groups (A-D) of clade 2.3.4.4.

 Specific remarks

The authors need to be discussed about epidemiological distribution of distinct groups (A-D) of clade 2.3.4.4.  

The manuscript needs careful rewriting to correct grammatical errors and some structural errors in the sentences .......

Page 4 line no 7: remove ‘a’ before multiple reassortments of the H10N5

Page 4 line no 10: add ‘a’ before few documented

Page 4, section 3.2. H5N8 (1st paragraph): With the launching of massive avian …………in Eastern China in 2010. I think it should be written

Author Response

Dear Reviewer:

We highly appreciate the reviews and comments our manuscript, entitled: “The Emergence and Decennary distribution of clade 2.3.4.4 HPAI H5Nx” have received. This manuscript was conceptualized on the idea of understanding the diversity and continuous emergence of 2.3.4.4 H5Nx viruses. 

Hence, we would like also to sincerely send our utmost gratitude to you as our reviewer, for the comments and suggestions given for the improvement of this paper. We have carefully re-checked and re-analyzed all the information indicated in this manuscript (maps, references and noted and added information). We hope that as we include our point by point responses for each comment/suggestions/clarification, we could clear and clarify all the information regarding this manuscript.

Thank you very much.

Best regards:

Khristine Joy Antigua

Comment/suggestion/Clarification

1.      For Validation:

a.      The authors need to be discussed about epidemiological distribution of distinct groups (A-D) of clade 2.3.4.4.  

Response: As suggested by the Reviewer, we have added in Section 3: Evolution of H5Nx viruses, a paragraph for discussion of the 2.3.4.4 provisional grouping. It reads as:

3.7 H5Nx Provisional Groupings

With the continuous emergence and observed diversity patterns of H5 viruses, Lee et al. proposed a provisional grouping for viruses belonging to clade 2.3.4.4. The H5Nx viruses were grouped into 4 (A-D) based on the phylogenetic relationships and the temporal evolutionary history of viruses when investigated using molecular clock analysis[10]. Moreover, Lee et al also have demonstrated that the genetic relatedness of viruses in each group are well-supported by the observed high bootstrap values (>70%) and long HA branches[9,10]. Group A or 2.3.4.4A is composed of viruses such as: a.) H5N8, identified from China in early 2014 and from South Korea, Japan, Taiwan, Canada, European countries and North America; b.) H5N1 and H5N2 from North America; c.) H5N2 and H5N3 from Taiwan. On the other hand, group B or 2.3.4.4B are primarily composed of H5N8 viruses identified in China in 2013 and South Korea in 2014[9,10]. Moreover, recent studies have suggested that H5N6 from South Korea in 2017-2018 as well as H5N5 and H5N8 from Germany in 2016-2017 also belong to this group by phylogenetic relatedness[79,80]. Group C of 2.3.4.4 clade were comprised of H5N6 from China and Laos in 2013 – 2014 and H5N1 identified from Vietnam and China in 2014[9,10]. Kim et al. also reported that H5N6 identified in Korea in 2016-2017 shares the phylogenetic relatedness in this group of 2.3.4.4[81]. Remarkedly, 2013-2014 H5N6 identified in China and Vietnam were grouped in 2.3.4.4D[9,10]. Though this provisional grouping remained unofficial however few studies have adapted the groupings to elucidate the diversity patterns of Clade 2.3.4.4 viruses or H5Nx based on molecular clock analysis.

b.      With the launching of massive avian …………in Eastern China in 2010. I think it should be written

Response: We carefully reviewed the statement and rephrase it. It now reads as:

“With the massive avian influenza surveillance and sample collection among domestic live poultry markets and farms in China, the novel subtype H5N8 (A/duck/Jiangsu/k1203/2010), along with the three H5N5 subtypes, was first detected in Eastern China in 2010 (Figure 1) [47].”

c.       Page 4 line no 7: remove ‘a’ before multiple reassortments of the H10N5; Page 4 line no 10: add ‘a’ before few documented

Response: We have corrected the unnecessary articles mentioned.